# Accessibility and allocation of public parks and gardens in England and Wales: A COVID-19 social distancing perspective

Niloofar Shoari[1], Majid Ezzati[1,2,3], Jill Baumgartner[4], Diego Malacarne[1], Daniela Fecht[1] *

**1** MRC Centre for Environment & Health, Department of Epidemiology and Biostatistics, School of Public Health, Imperial College London, London, United Kingdom, **2** Abdul Latif Jameel Institute for Disease and Emergency Analytics, Imperial College London, London, United Kingdom, **3** Regional Institute for Population Studies, University of Ghana, Legon, Ghana, **4** Institute for Health and Social Policy and Department of Epidemiology, Biostatistics and Occupational Health, McGill University, Montreal, Quebec, Canada

* d.fecht@imperial.ac.uk

## Abstract

Visiting parks and gardens supports physical and mental health. We quantified access to public parks and gardens in urban areas of England and Wales, and the potential for park crowdedness during periods of high use. We combined data from the Office for National Statistics and Ordnance Survey to quantify (i) the number of parks within 500 and 1,000 metres of urban postcodes (i.e., availability), (ii) the distance of postcodes to the nearest park (i.e., accessibility), and (iii) per-capita space in each park for people living within 1,000m. We examined variability by city and share of flats. Around 25.4 million people (~87%) can access public parks or gardens within a ten-minute walk, while 3.8 million residents (~13%) live farther away; of these 21% are children and 13% are elderly. Areas with a higher share of flats on average are closer to a park but people living in these areas visit parks that are potentially overcrowded during periods of high use. Such disparity in urban areas of England and Wales becomes particularly evident during COVID-19 pandemic and lockdown when local parks, the only available out-of-home space option, hinder social distancing requirements. Cities aiming to facilitate social distancing while keeping public green spaces safe might require implementing measures such as dedicated park times for different age groups or entry allocation systems that, combined with smartphone apps or drones, can monitor and manage the total number of people using the park.

## Introduction

Public parks and gardens, being the most visited form of green space among urban residents in the UK [1], contribute to wellbeing by providing opportunities to experience nature, engage in physical activity, and feel a sense of social belonging and develop social interactions [2–7]. There are an increasing number of studies that explore health benefits of parks using various measures based on spatial proximity [8–10], density (number of parks within a certain

**Data Availability Statement:** The green space data were acquired from https://www.ordnancesurvey.co.uk/business-government/products/open-map-greenspace. The population data were obtained

from https://www.nomisweb.co.uk/census/2011/postcode_headcounts_and_household_estimates and https://www.ons.gov.uk/peoplepopulationandcommunity/populationandmigration/populationestimates/datasets/lowersuperoutputareamidyearpopulationestimates.

**Funding:** This work is supported by the Pathways to Equitable Healthy Cities grant from the Wellcome Trust [209376/Z/17/Z]. Infrastructure support for the Department of Epidemiology and Biostatistics was provided by the NIHR Imperial Biomedical Research Centre (BRC). The funders did not have any input in study design; in the collection, analysis and interpretation of data; in the writing of the report; and in the decision to submit the paper for publications. The corresponding author had full access to all the data in the study and had final responsibility for the decision to submit for publication.

**Competing interests:** The authors have read the journal's policy and have the following competing interests: ME received a charitable grant from the AstraZeneca Young Health Programme and personal fees from Prudential, outside the submitted work. This does not alter our adherence to PLOS ONE policies on sharing data and materials. There are no patents, products in development or marketed products associated with this research to declare.

distance from home or any other population boundary) [9, 11, 12], crowdedness [13, 14], quality and facilities [15–17], or a combination of the above [18].

The lockdown periods during the COVID-19 pandemic have made the crucial role of natural spaces even more evident and exposed the limited resources in our cities that led to diverse policy and advocacy responses in the UK and other parts of the world, ranging from park closures to limiting park opening times and reducing services such as park benches, children's play areas, and sports facilities [19, 20]. Research has showed that extended periods of confinement at home reduce physical activity, particularly among people with lower socio-economic status [21], and increase the risk of depression, anxiety, insomnia, and self-harm [22, 23]. Concerns were also raised that such unintended consequences disproportionately affected children and disadvantaged communities living in overcrowded homes and inner-city flats without access to outdoor space or private gardens [24–27].

While there have been studies on access and provision of public parks and gardens at city-level in the UK, there is a knowledge gap at national level with a focus on within- and between-city variabilities. Here, we describe the availability, accessibility, and provision of publicly owned parks and gardens in urban areas in England and Wales, and consider policy options for their allocation and use.

## Methods

### Data sources

Our analysis focused on urban areas in England and the three most populous cities in Wales (i.e., Cardiff, Swansea, and Newport). We focused on urban areas because those living in rural areas typically have private gardens and/or access to the countryside. The urban areas in England were defined using the Built-up Areas boundaries from the Office for National Statistics (ONS) [28] and in Wales using the boundary data provided by local authorities.

We used age-stratified ONS mid-2018 population estimates at Lower Super Output Areas (LSOA), a census dissemination unit which represents homogeneous neighbourhoods of 1,500 residents on average. LSOA-level population was matched to residential postcodes centroids (representing on average 15 households in urban areas) using postcode headcount information from the 2011 UK Census as weights. We aggregated populations into five age categories: children and young adolescents (0–16 years), young adults (16–30 years), middle-aged adults (31–50 years and 51–70 years), and the elderly (70+ years). We obtained information on type of accommodation (residential flat versus house) from the 2011 UK Census for LSOAs. We identified public parks and gardens using the OS MasterMap Open Greenspace Layer (version October 2019), which provides information on the location, physical boundary, and function of publicly accessible green space.

### Statistical analysis

We used a geographic information system (GIS) to conduct the following analyses: First, we quantified the availability of public green space, defined as total number of parks and gardens within 500 and 1,000 metres circular buffers around all residential postcodes (referred to as availability). The 500 and 1,000 metres buffer sizes approximately represent five and ten minutes of walking for an adult, respectively [29]. Second, we quantified the accessibility of green space, measured by spatial proximity, using the (Euclidean) distance between the postcode centroid and the nearest public park or garden. Third, we quantified the per-capita space available in each public park and garden by dividing its total area by the population size residing within 1,000 metres. Since park usage data were not available, we estimated the population that might be expected in public parks and gardens based on the following scenarios:

- Scenario A: population size was weighted based on the number of parks within a 1,000 metres circular buffer around each postcode. This scenario assumes that all public parks within the buffer have an equal chance of being visited [13]. For example, if a given postcode had three parks available within its 1,000 metres buffer, 33% of population was assigned to each park.

- Scenario B: We used a simplified spatial interaction model to calculate population weights based on the probability of visiting a given park or garden within 1,000 meters of each post-code. This scenario assumes that parks that are larger and closer to a given urban postcode are more likely to be visited [18]. Mathematically, the weight function is expressed as

$$W_{ij} = \frac{A_j/d_{ij}^{\alpha}}{\sum A_j/d_{ij}^{\alpha}}$$

where population weights ($W_{ij}$) are defined as the probability that people living at postcode $i$ visits park $j$, $A_j$ is the area of park $j$, $d_{ij}$ is the distance between postcode $i$ and park $j$, and $\alpha$ is the distance decay parameter, which based on previous studies was assumed equal to 2.0 [30, 31]. We used the per-capita space measure as an indicator of the possible crowdedness of the park or garden, which is also useful in evaluating the feasibility of urban parks and gardens to facilitate social distancing.

Finally, we examined accessibility and mean per-capita space of parks and gardens in relation to the proportion of homes in an LSOA that are flats, where a higher proportion of flats indicates greater reliance on public parks and gardens for green space access [24]. We conducted the analyses in ArcMap v.10.5.1 (ESRI Ltd, Redlands, California) and R Statistical Software (Version 1.2.5001).

## Results

Our analysis covered 537,713 urban postcodes in England and Wales, with a total population of over 29 million. Of these, ~6.2 million (21%) were children and young adolescents (0–16 years), ~5.9 million (20%) young adults (16–30 years), ~14.3 million (49%) middle-aged adults (31–50 years and 51–70 years), and 2.9 million (10%) 70+ years of age. There were a total of 4,155 public parks and gardens in urban areas in England and Wales.

### Availability and accessibility of public parks and gardens

There is on average one (standard deviation [SD] 1.2) public park and garden available within 500 metres of the urban postcodes in England and Wales, and three (SD 2.8) public parks and garden within 1,000 metres. Forty-three percent of postcodes in England and Wales do not have any public green space within 500 metres, 34% have one, and 23% have two or more. Fourteen percent and 22% of postcodes have either no park or one park, respectively, within 1,000 metres whereas 63% of postcodes have at least two parks and gardens.

Urban residents in England and Wales, on average, live 557 metres away from their closest park or garden. Ten percent of the population (2.8 million) has at least one park in the immediate vicinity of their residence (< 100 metres), 59% (14.4 million) within a 500 metres, 28% (8.3 million) between 500 to 1,000 metres, and 13% (3.8 million) live more than 1,000 metres from a public park or garden.

There are substantial differences in distance to public green space between cities (Fig 1). Bristol, Liverpool, and London have the best accessibility with median distances of 281 metres, 284 metres, and 322 metres, respectively. By comparison, Newport (median distance of 673 metres), Swansea (611 metres), Coventry (522 metres), and Leeds (518 metres) are cities with

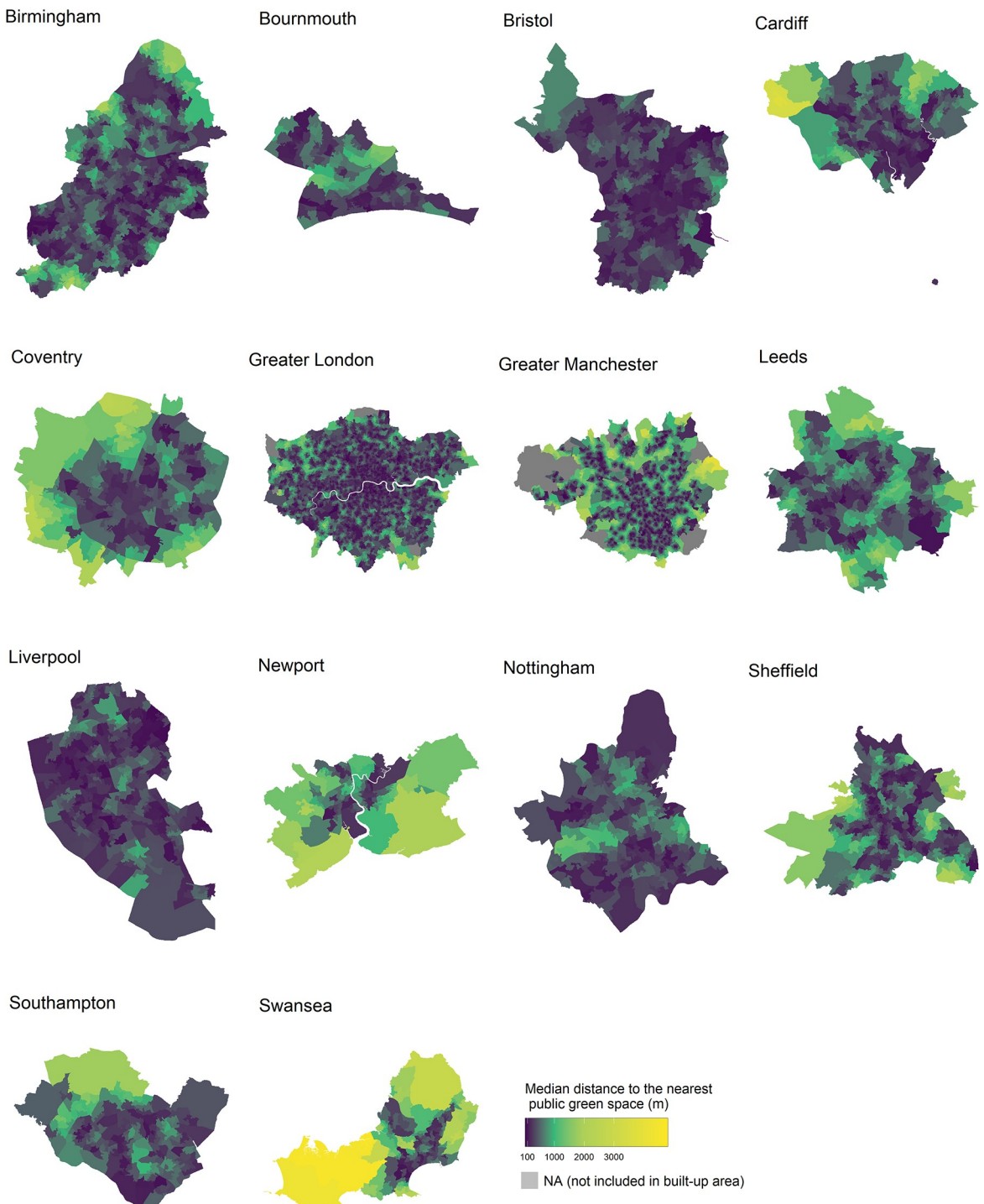

**Fig 1. Median distance between home postcode and the nearest public park or garden in Lower Super Output Areas (LSOA) for 14 cities in England and Wales.** Contains National Statistics data © Crown copyright and database right 2020. Contains OS data © Crown copyright (2020). Data available under the UK Open Government Licence v3.

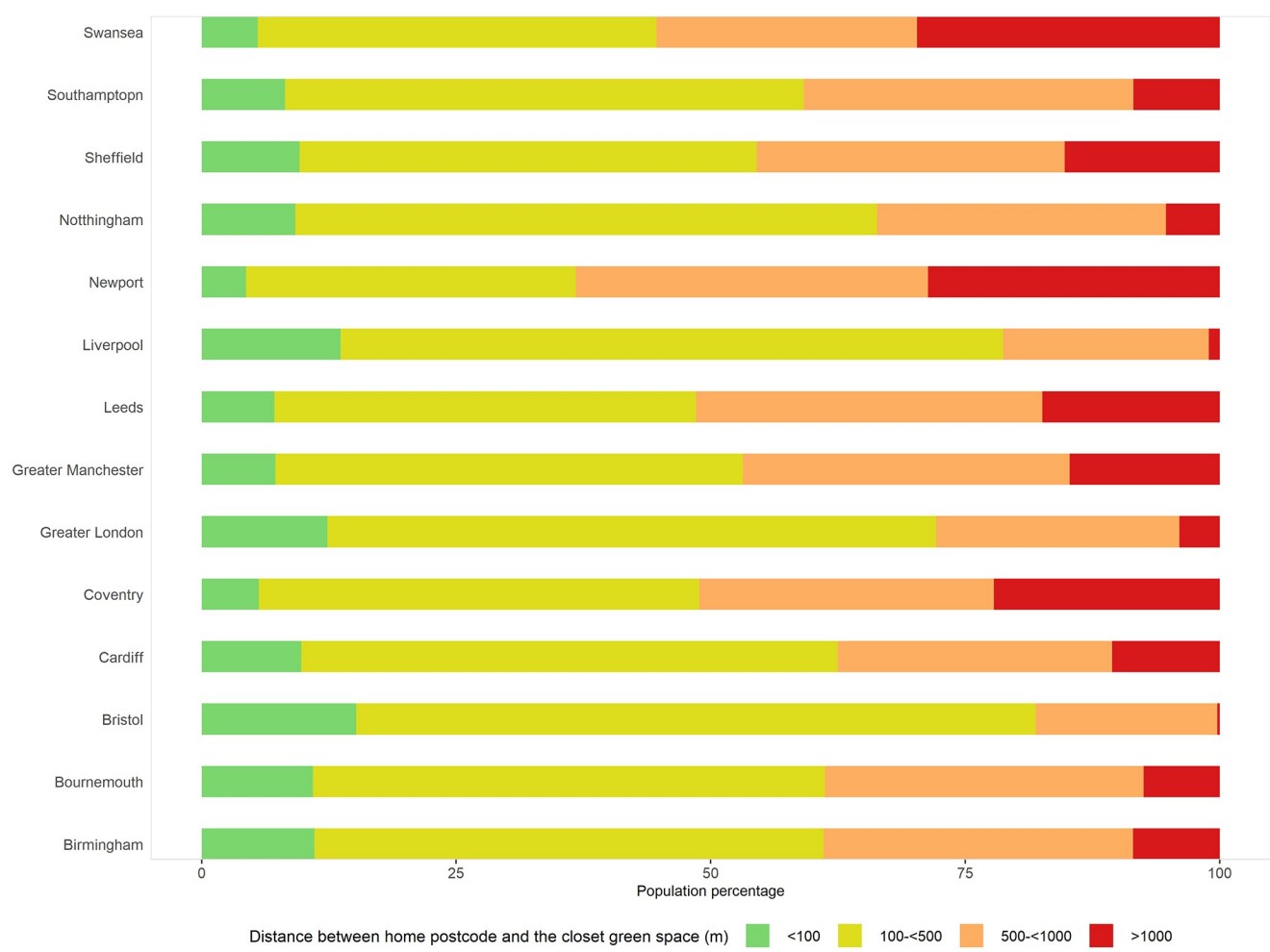

**Fig 2. Percent of population in categories of distance to the nearest park or garden for 14 cities in England and Wales.**

the least accessibility. In terms of the percentage of population, ~5% of people in Swansea and Coventry and 7% in Leeds and Manchester have a public park or garden in the immediate vicinity (< 100 metres) of their residence compared with 15% in Bristol, 14% in Liverpool and 12% in London (Fig 2). The former four cities also have the highest percentages of population with a park located more than 1,000 metres from their residence, ranging from 15% in Manchester to 30% in Swansea, compared with just 0.2% in Bristol, 1% in Liverpool, and 4% in London.

## Per-capita space of public parks and gardens

Fig 3 shows the distribution of per-capita green space for all people (25.4 million) living within 1,000 metres of public parks and gardens in urban areas of England and Wales. To evaluate whether this available green space allows maintaining a distance of at least two metres for social distancing, a minimum space of four square meters per person is required if people were spread evenly within the park. If people concentrate in certain areas of the park, such as paths, the requirement is higher. At the extreme, if urban residents were all to visit their closest park at the same time, 50% of parks (2,071) would be unable to maintain the minimum social

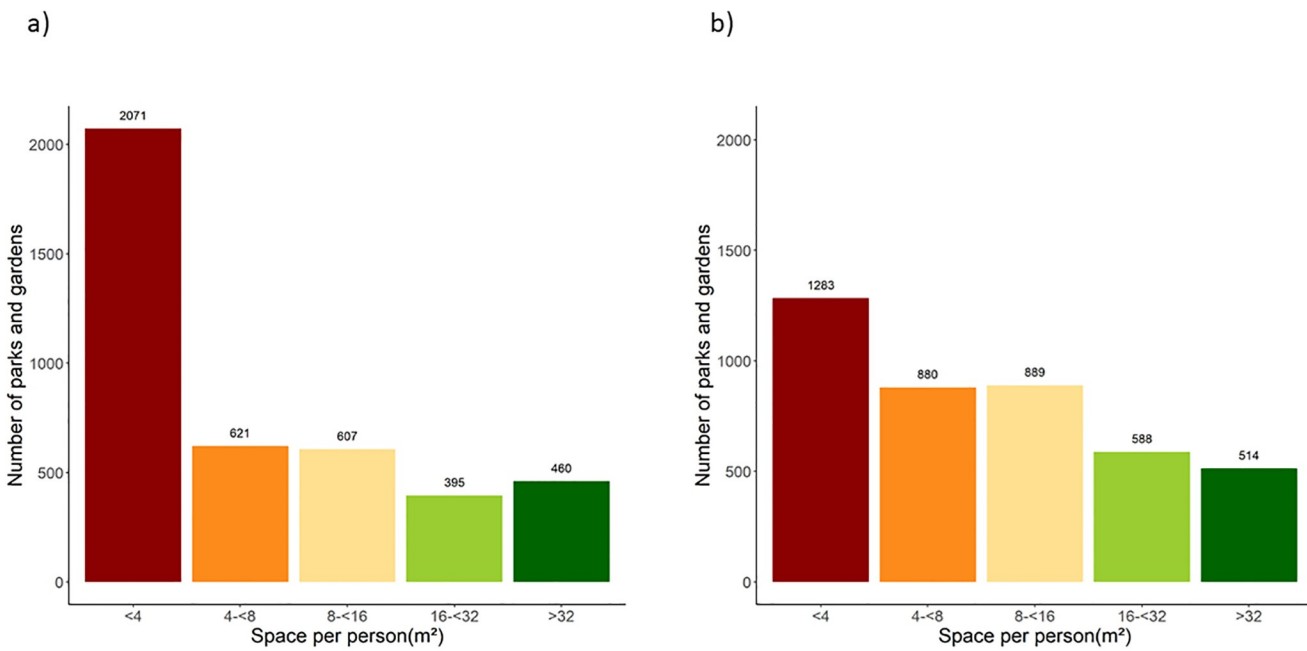

**Fig 3. Distribution of space per person for public parks and garden in England and Wales for a) scenario A (all parks within 1,000 meters have an equal chance of being visited), and b) scenario B (larger and closer parks to home are more likely to be visited).**

distancing space of four square metres per person, even if the entire park space was used. In an alternative scenario, where urban residents choice of parks depends on the size of park and its distance to their home (i.e., scenario B as described in Statistical Analysis), an additional 19% of parks and gardens (69% total) would enable social distancing as long as people were equally spread out in the park.

For over 7 million of urbanites in England and Wales who have access to parks or gardens within 1,000 metres from their homes, there is a risk of going to a park (if all residents use the park simultaneously) that is potentially overcrowded with less than four square meter of green space per person (Fig 4). Of this group, 21% are children and young adolescents, and 9% are elderly. Around 3.8 million people do not have park or garden within 1,000 metres of their homes, of which 21% and 13% are younger than 16 years and 70 years and older, respectively.

LSOAs with a higher share of flats generally have a better accessibility to parks based on distance (Fig 5a), though parks in these LSOAs are more likely to be overcrowded if used by all residents (Fig 5b). For example, residents in the highest quantile of share of flats can reach a park or garden within 278 metres (the median) but the median space available in all parks in their LSOA is as small as 4.9 square metres per person, which is 2.5 times smaller than the per-capita space available in the lowest quantile of share of flats.

## Discussion

Cities in England and Wales have green space assets that provide opportunities for outdoor exercise and play, but there are bottlenecks for some urbanites. Specifically, 13% of residents live more than 1,000 metres (~ten minutes walk) from their nearest public park or garden. Among those with a good access to local parks, 24% of residents use parks that are potentially overcrowded with as little as four square meter of green space per person. This makes the park experience less than desirable and, during a pandemic, unsafe to maintain social distancing.

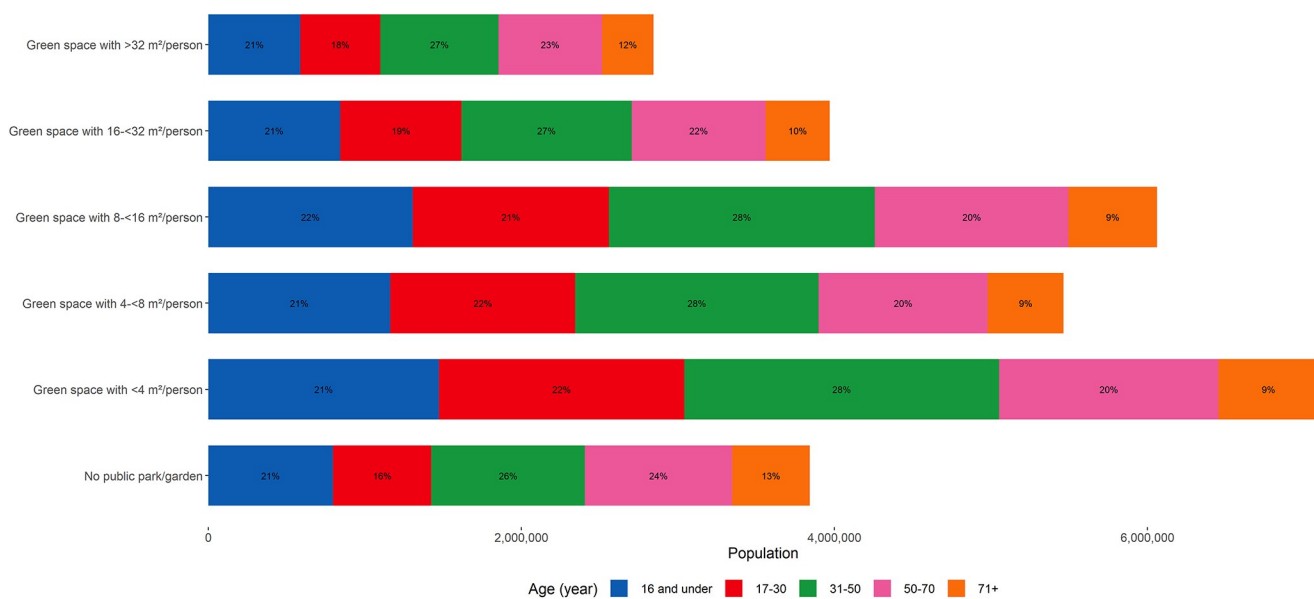

**Fig 4. Number of people in each category of park space availability and age group.** Numbers refer to the people living within 1,000 metres of any park or garden.

The main strength of our study is to provide data on green space availability and accessibility as a national assessment tool to identify areas with high demand for public green space. Additionally, it can inform options to keep public parks and gardens open and safe during lockdown periods. A limitation of our work is that we did not consider other spaces (e.g. national parks and woodlands) although these are usually not located in urban areas. Further,

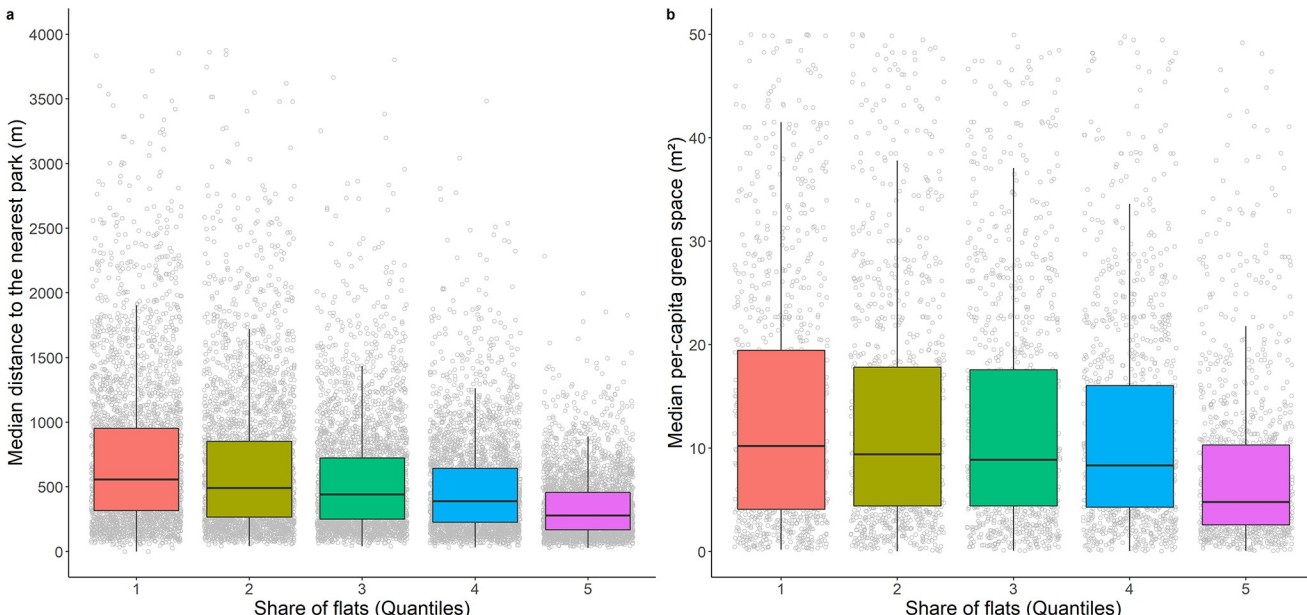

**Fig 5. a) Median distance (m) to the nearest park or garden by quintiles of share of flats in Lower Super Output Area; and b) Median per-capita space (m²) by share of flats in Lower Super Output Area (quantiles), where 1 is the smallest proportion of flats and 5 is the highest proportion of flats.** In Fig 5b, the y-axis was delimited to 0 m² and 50 m² for presentation purposes.

our analysis could provide more detailed insights on crowding if there was information on time preferences for exercise and play, and attributes of parks beyond their amenities and services [17]. Since the data source for public parks and gardens provide information only on the form and function of green space, we did not subtracted inaccessible areas such as bodies of water and monuments from the total park area.

## How to keep public parks and gardens accessible and safe during COVID-19 pandemic?

Public parks and gardens provide urban residents of all ages with access to outdoor green space and a place to exercise, which can reduce stress and improve mental and physical health [32, 33]. As social distancing measures are prolonged in the absence of a vaccine or widespread test and contact tracing, local and national governments will have to balance the access to outdoor green space with reducing the risk of transmission, particularly in densely populated areas. The Paris model of closing all parks reduces social contact in public spaces [34], but can have serious mental and physical health implications. Further, these more extreme measures may cause other outdoor spaces to become more crowded, risk engendering non-compliance with social distancing regulations, and can create tensions among residents and with officials that enforce the regulations.

An alternative policy is restricting access to high-risk areas (e.g., playgrounds and sports facilities) while keeping trails and open spaces accessible in a way that maintains social distancing. For example, parks can limit the number of people accessing based on park size and population density in the surrounding area [35]. Dedicated park access times for different age groups or different activities could serve to both maintain social distancing and facilitate access for more vulnerable groups. Examples include specific times for families and the elderly and for walkers versus runners and cyclists. Alternatively, officials could manage utilisation, either based on weekly data to inform the community so that they can better spread their visits over the park's opening times, or dynamically using smartphone or drone data to monitor crowdedness and communicate this information to residents.

Finally, given the extreme nature of the pandemic as a social and public health crisis, cities should complement public parks and gardens with other resources to lessen the adverse impacts of lockdown and social distancing. In Boston, Minneapolis and Oakland in United States, cities closed streets to vehicles to increase space for pedestrian and cyclists [36]. Similarly, in the UK, some local authorities in London, Manchester, and Brighton are restricting driving on certain roads to separate walkers from runners and cyclists [37]. Coordinating such an initiative would allow for longer routes and safer activities, provide alternative spaces for different activities (e.g., adult cyclists versus playing/running children), and potentially reduce congestion in parks. Opening up school green land, private parkland and golf courses to the public can provide additional space for exercising while maintaining social distancing [27]. For example, Dulwich College and Dulwich Prep in London have opened up sections of their land to the public.

While a great deal of our attention in the early months of the pandemic is on supressing or stopping transmission, the strategies for achieving this can have detrimental impacts on health and wellbeing. Public parks and gardens are an important public health asset that can effectively help urban population to sustain their health and wellbeing, and should actively and effectively be used to do so.

## Author Contributions

**Conceptualization:** Majid Ezzati.

**Data curation:** Niloofar Shoari.

**Formal analysis:** Niloofar Shoari.

**Funding acquisition:** Majid Ezzati.

**Methodology:** Niloofar Shoari, Majid Ezzati, Jill Baumgartner, Daniela Fecht.

**Supervision:** Daniela Fecht.

**Visualization:** Niloofar Shoari, Diego Malacarne.

**Writing – original draft:** Niloofar Shoari, Majid Ezzati, Jill Baumgartner, Daniela Fecht.

**Writing – review & editing:** Niloofar Shoari, Majid Ezzati, Jill Baumgartner, Daniela Fecht.

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
