## [Decision Letter · Decision Letter 0]

28 Jul 2020

PONE-D-20-13318

Accessibility and allocation of public parks and gardens during COVID-19 social distancing in England and Wales

PLOS ONE

Dear Dr. Fecht,

Thank you for submitting your manuscript to PLOS ONE. After careful consideration, we feel that it has merit but does not fully meet PLOS ONE’s publication criteria as it currently stands. Therefore, we invite you to submit a revised version of the manuscript that addresses the points raised during the review process.

My sincere apologies for the time taken to review your submission- as you can appreciate it is hard to secure reviewers in these difficult times. Please do take the reviewers comments on board- both have expressed the utility of the findings generally, and not specifically for COVID-19.

We look forward to receiving your revised manuscript.

Kind regards,

Sreeram V. Ramagopalan

Academic Editor

PLOS ONE

Journal Requirements:

2.We note that [Figure(s) 1] in your submission contain [map/satellite] images which may be copyrighted. All PLOS content is published under the Creative Commons Attribution License (CC BY 4.0), which means that the manuscript, images, and Supporting Information files will be freely available online, and any third party is permitted to access, download, copy, distribute, and use these materials in any way, even commercially, with proper attribution. For these reasons, we cannot publish previously copyrighted maps or satellite images created using proprietary data, such as Google software (Google Maps, Street View, and Earth). For more information, see our copyright guidelines: http://journals.plos.org/plosone/s/licenses-and-copyright.

1.    You may seek permission from the original copyright holder of Figure(s) [1] to publish the content specifically under the CC BY 4.0 license. 

Reviewers' comments:

Reviewer's Responses to Questions

**Comments to the Author**

1. Is the manuscript technically sound, and do the data support the conclusions?

Reviewer #1: Partly

Reviewer #2: Yes

2. Has the statistical analysis been performed appropriately and rigorously? 

Reviewer #1: I Don't Know

Reviewer #2: Yes

3. Have the authors made all data underlying the findings in their manuscript fully available?

Reviewer #1: Yes

Reviewer #2: Yes

4. Is the manuscript presented in an intelligible fashion and written in standard English?

Reviewer #1: Yes

Reviewer #2: Yes

5. Review Comments to the Author

Reviewer #1: Access to green space is important for physical and mental health at all times, and I appreciate that you have chosen to address this issue. No doubt it is even more important during the restrictions imposed because of the Covid-19 epidemic, but I am concerned that your framing suggests the issue is important only under recent circumstances. Your title and introduction do not merely imply, but explicitly state that the observations are specific to the Covid-19 epidemic, which is not the case. Your title would be more accurate if it were shortened to Accessibility and allocation of public parks and gardens in England and Wales, since these observations are unaffected by the pandemic. You can certainly mention that the pandemic makes the question even more exigent than usual, but you don’t want to suggest that it will go away with the lockdown. The first sentence of your abstract reinforces this framing: “Visiting parks and gardens may attenuate the adverse physical and mental health impacts of 16 social distancing implemented to reduce the spread of COVID-19.” In fact visiting parks and gardens may attenuate the adverse impacts of all psycho-social stressors and one might say that lack of absence to green space is a stressor under any circumstances. The first sentence of your introduction continues this misleading framing.

That said, I have some clarifying questions and I also see some substantial limitations to this that need greater acknowledgment and I think you need to make a more substantive case for your procedures.

First, we already know that some people have greater access to open space than others. What specifically is this telling us that is useful for policy making or public health practice?

Now, some more specific comments.

Line 79: You need to explain what Lower Super Output Area means, especially as you return to the concept which will be unfamiliar to most readers. International readers will need to know what British postal codes are, since 15 households is considerably smaller than the granularity of postal codes elsewhere. Readers will also need to know how this database protects privacy, since out of 15 households, with the precise age data that is apparently available, individuals could evidently be readily identified. Again, in the U.S. census blocks vary widely but may have several hundred inhabitants, and any cells with fewer than 5 are suppressed.

Line 96: What is the typical geographic extent of a postcode? This will inform us as to the accuracy of using the centroids.

Line 100 et seq. This is very confusing. Did you consider the size of the parks at all? Assigning 33% of the population to one of three available parks makes no sense if they are of different sizes. You write: “minimum space of four square meters per person is required to maintain a distance of at least two metres if people were spread evenly within the park. If people concentrate in certain areas of the park, such as paths, the requirement is higher.” Well yes, but you do not explain whether you used the size of the parks in your calculations or if so how. Furthermore, regardless of whether people are concentrated on walkways, the available space may be reduced by bodies of water, shrubbery or garden beds, monuments, etc. How did you compute the actual space available per person, if at all? And if not, how are your calculations of possible overcrowding at all valid? There is basic information missing here.

Line 148: There appears to be at least one word missing. Also, you need to explain what a LSOA is.

Line 155 et seq: neither of your hypotheticals -- everybody visiting at the same time, or people evenly spaced – will ever be met. So what is the use of these calculations? (And again, it isn’t clear to me that you have actually measured the space available to begin with.)

Line 198: Well yes but you do not have any of this information.

So in sum, I need to better understand what you did, and you need to make a better case for its usefulness. I hope you can do so because this is certainly a very important issue that doesn’t get sufficient attention.

Reviewer #2: The topic of open space in the form of parks is interesting and of course relevant, especially in light of covid. That said, I wish the framing of the study were more general, not focused on covid specifically but the availability of and access to parks and other open space. There are several places where the focus on covid limits the generalizability and may date the findings (such as stay-at-home-orders expiring in May). Instead I think a broader focus on the health benefits of open spaces would be better, and some of the situations raised in the discussion used to frame this study.

With regard to the methods, this is a nice compilation of data. Parks are indeed and important opportunities for open space, but I wonder about those people especially in the fringes of the cities have other opportunities to walk to open areas. Perhaps this could account for why some of the smaller cities seem more problematic than London, especially Newport and Coventry. The spacial maps appear to support this. This does not appear to be the case in Liverpool and Nottingham, although this may be because there are outdoor places along the fringes of these cities. Although I would ideally like this factor of alternative open spaces included and analyzed, at the very least this could be acknowledged as a limitation to this study.

6. PLOS authors have the option to publish the peer review history of their article (what does this mean?). If published, this will include your full peer review and any attached files.

Reviewer #1: **Yes: **M. Barton Laws, Ph.D.

Reviewer #2: No

---

## [Author Response · Author response to Decision Letter 0]

8 Oct 2020

Response to Journal:

J1 Please ensure that your manuscript meets PLOS ONE's style requirements, including those for file naming. 

RJ1 We have carefully assessed the manuscript and made the necessary changes to fully comply with the journal’s style requirements. 

J2 We note that [Figure(s) 1] in your submission contain [map/satellite] images which may be copyrighted. 

RJ2 Figure 1 uses information from the Office for National Statistics and Ordnance Survey which is freely available under the UK Open Government Licence v3. To use and reproduce the data a copyright statement needs to be added to the map. We have now included this statement in the Figure legend as follows: Contains National Statistics data © Crown copyright and database right 2020. Contains OS data © Crown copyright (2020). Data available under the UK Open Government Licence v3.

Response to Reviewer:

Reviewer #1

R1.1 Access to green space is important for physical and mental health at all times, and I appreciate that you have chosen to address this issue. No doubt it is even more important during the restrictions imposed because of the Covid-19 epidemic, but I am concerned that your framing suggests the issue is important only under recent circumstances. Your title and introduction do not merely imply, but explicitly state that the observations are specific to the Covid-19 epidemic, which is not the case. Your title would be more accurate if it were shortened to Accessibility and allocation of public parks and gardens in England and Wales, since these observations are unaffected by the pandemic. You can certainly mention that the pandemic makes the question even more exigent than usual, but you don’t want to suggest that it will go away with the lockdown. The first sentence of your abstract reinforces this framing: “Visiting parks and gardens may attenuate the adverse physical and mental health impacts of 16 social distancing implemented to reduce the spread of COVID-19.” In fact visiting parks and gardens may attenuate the adverse impacts of all psycho-social stressors and one might say that lack of absence to green space is a stressor under any circumstances. The first sentence of your introduction continues this misleading framing.

RR1.1 We fully agree with the reviewer that access to parks is important under all circumstances, not just during COVID-19 restrictions. We followed the reviewer’s advice and shortened the title to ‘Accessibility and allocation of public parks and gardens in England and Wales’ but made it clear that we are assessing this through the COVID-19 lens by adding ‘a COVID-19 social distancing perspective’. We have made structural changes throughout the paper to better reflect the importance of parks and gardens at all times, and especially during the COVID-19 pandemic.

R1.2 That said, I have some clarifying questions and I also see some substantial limitations to this that need greater acknowledgment and I think you need to make a more substantive case for your procedures. First, we already know that some people have greater access to open space than others. What specifically is this telling us that is useful for policy making or public health practice?

RR1.2 From a public health perspective, the number of parks, their distance to residential postcodes, and per-capita green space provided to local residents represent not only spatial accessibility to parks but also indicate disparities in the quality of park visits. The covid-19 restrictions made it more clear that parks are not a “good to have” option but a necessity to physical and mental health. The importance of this kind of analysis is clear from the fact that during lockdown people were bound to homes and the only outside-home options were parks and gardens. In addition, when lockdown measures were announced, concerns were raised regarding the need to accompany lockdown requirements with measures to create access to parks. Another advantage is that our study at a national level enabled us to perform comparable analyses over a large number of urban areas and observe patterns in different cities. From a policy-making perspective, our findings lay the foundation to inform future needs for alternative public green space such as pocket parks or even the need for new parks in some neighbourhoods, especially those with higher share of flats who are most in need of public green space.

R1.3 Line 79: You need to explain what Lower Super Output Area means, especially as you return to the concept which will be unfamiliar to most readers. International readers will need to know what British postal codes are, since 15 households is considerably smaller than the granularity of postal codes elsewhere. Readers will also need to know how this database protects privacy, since out of 15 households, with the precise age data that is apparently available, individuals could evidently be readily identified. Again, in the U.S. census blocks vary widely but may have several hundred inhabitants, and any cells with fewer than 5 are suppressed.

RR1.3 We added explanations regarding the postcode and LSOA geography in the UK in the text as follows: ‘LSOAs […] a census dissemination unit which represents homogeneous neighbourhoods of 1,500 residents on average’ and ‘…residential postcode centroids (representing on average 15 households in urban areas…’. The UK postcode system is different from that of other countries such as the US in that postcodes do not represent an area but a collection of postal delivery points (i.e. addresses) which are geographically represented by a postcode centroid (x,y coordinate). Postcode headcount information which was used here to disaggregate the LSOA level population estimates comes from the national census and gives the total number of people sharing the same postcode on that date of the census. No data suppression is applied to postcode headcount by the data provider, Office for National Statistics.

R1.4 Line 96: What is the typical geographic extent of a postcode? This will inform us as to the accuracy of using the centroids.

RR1.4 As outlined in response 1.3, postcodes are the midpoint (x,y location) of a collection of postal delivery points (~15 addresses) sharing the same postcode. 

R1.5 Line 100 et seq. This is very confusing. Did you consider the size of the parks at all? Assigning 33% of the population to one of three available parks makes no sense if they are of different sizes. You write: “minimum space of four square meters per person is required to maintain a distance of at least two metres if people were spread evenly within the park. If people concentrate in certain areas of the park, such as paths, the requirement is higher.” Well yes, but you do not explain whether you used the size of the parks in your calculations or if so how. Furthermore, regardless of whether people are concentrated on walkways, the available space may be reduced by bodies of water, shrubbery or garden beds, monuments, etc. How did you compute the actual space available per person, if at all? And if not, how are your calculations of possible overcrowding at all valid? There is basic information missing here.

RR1.5 We did use the size of park in calculating the space available but not as a determinant of preference to use. To our knowledge there is no empirical evidence on this. In other words, we assumed that all parks have the same chance of being visited. We have now added citations that used this approach to estimate available green space. In the revised version, we took into account how the size of a park and the distance to a residential postcode might affect the probability of visiting a given park. We calculated population weights assuming that people tend to visit larger parks that are closer to their home. Relevant citations have been also added to the revised version. Regarding Reviewers concern on the actual available green space in parks, the park data provided information on green space with the function of park and garden as a whole. Currently, we are not aware of any data source containing parks details and facilities (such as bodies of waters and monuments) at a national scale. This lack of data was acknowledged as a limitation in the revised version 

R1.6 Line 148: There appears to be at least one word missing. Also, you need to explain what a LSOA is.

RR1.6 We have revised as suggested and include an explanation of LSOAs ‘LSOAs […] a census dissemination unit which represents homogeneous neighbourhoods of 1,500 residents on average’.

R1.7 Line 155 et seq: neither of your hypotheticals -- everybody visiting at the same time, or people evenly spaced – will ever be met. So what is the use of these calculations? (And again, it isn’t clear to me that you have actually measured the space available to begin with.)

RR1.7 In the absence of park usage data, we had to make assumption on how to assign populations to parks to be able to estimate the provision of green space. Previous studies showed that green space characteristics such as size, quality, facilities, and distance contribute to usage preferences. Therefore, we have refined our methods based on an alternative assumption that residents divide between competing parks within their ~10 min walk depending on the size of park and the distance to their homes. Data collection on frequency of visits and high-use times will definitely improve the accuracy of results; however, it goes beyond the purpose of this research.

R1.8 Line 198: Well yes but you do not have any of this information. So in sum, I need to better understand what you did, and you need to make a better case for its usefulness. I hope you can do so because this is certainly a very important issue that doesn’t get sufficient attention.

RR1.8 Although this information is not currently available, we believe that our findings provide valuable insight on the current accessibility and provision of parks and gardens in the UK and Wales. We acknowledged as a limitation that due to unavailability of detailed data on attributes of parks such as facilities, cafes, etc. we were unable to calculate the exact amount of usable park space. 

Reviewer #2: 

R2.1 The topic of open space in the form of parks is interesting and of course relevant, especially in light of covid. That said, I wish the framing of the study were more general, not focused on covid specifically but the availability of and access to parks and other open space. There are several places where the focus on covid limits the generalizability and may date the findings (such as stay-at-home-orders expiring in May). Instead I think a broader focus on the health benefits of open spaces would be better, and some of the situations raised in the discussion used to frame this study.

RR.2.1 As highlighted by the Reviewer, access to parks is important under all circumstances but became more noticeable during COVID-19 restrictions. To reflect the reviewer’s comment (also in response to Reviewer #1) we changed the title to ‘Accessibility and allocation of public parks and gardens in England and Wales’ and made it clear that we are assessing this through the COVID-19 lens by adding ‘a COVID-19 social distancing perspective’. We have made structural changes throughout the paper to better reflect the importance of parks and gardens at all times, and especially during the COVID-19 pandemic.

R2.2 With regard to the methods, this is a nice compilation of data. Parks are indeed and important opportunities for open space, but I wonder about those people especially in the fringes of the cities have other opportunities to walk to open areas. Perhaps this could account for why some of the smaller cities seem more problematic than London, especially Newport and Coventry. The spacial maps appear to support this. This does not appear to be the case in Liverpool and Nottingham, although this may be because there are outdoor places along the fringes of these cities. Although I would ideally like this factor of alternative open spaces included and analyzed, at the very least this could be acknowledged as a limitation to this study.

RR2.2 The difference of access to green space between small and large cities is related to the differences in cities’ morphology and distribution of public parks. For example, Nottingham is mostly characterised by urban zones with public parks distributed across the city while Newport is characterised by a mixed morphology of urban- rural areas, with urban areas and parks in the city centre. We have pointed out in line 216 that we just considered public parks and gardens and not other open spaces such as woodlands or national parks.

---

## [Editor Report · Decision Letter 1]

9 Oct 2020

Accessibility and allocation of public parks and gardens in England and Wales: a COVID-19 social distancing perspective

PONE-D-20-13318R1

Dear Dr. Fecht,

We’re pleased to inform you that your manuscript has been judged scientifically suitable for publication and will be formally accepted for publication once it meets all outstanding technical requirements.

Kind regards,

Sreeram V. Ramagopalan

Academic Editor

PLOS ONE
---

## [Editor Report · Acceptance letter]

16 Oct 2020

PONE-D-20-13318R1 

Accessibility and allocation of public parks and gardens in England and Wales: a COVID-19 social distancing perspective 

Dear Dr. Fecht:

I'm pleased to inform you that your manuscript has been deemed suitable for publication in PLOS ONE. Congratulations! Your manuscript is now with our production department. 

Kind regards, 

on behalf of

Dr. Sreeram V. Ramagopalan 

Academic Editor

PLOS ONE